



# Continued Results from a Field Campaign of Wake Steering Applied at a Commercial Wind Farm: Part 2

Paul Fleming[1], Jennifer King[1], Eric Simley[1], Jason Roadman[1], Andrew Scholbrock[1], Patrick Murphy[1,3], Julie K. Lundquist[1,3], Patrick Moriarty[1], Katherine Fleming[1], Jeroen van Dam[1], Christopher Bay[1], Rafael Mudafort[1], David Jager[1], Jason Skopek[2], Michael Scott[2], Brady Ryan[2], Charles Guernsey[2], and Dan Brake[2]

[1]National Wind Technology Center, National Renewable Energy Laboratory, Golden, CO, 80401, USA
[2]NextEra Energy Resources, 700 Universe Blvd, Juno Beach, FL, 33408
[3]Dept. Atmospheric and Oceanic Sciences, University of Colorado Boulder, Boulder, CO, 80303, USA

*Correspondence to:* Paul Fleming (paul.fleming@nrel.gov)

**Abstract.** This paper presents the results of a field campaign investigating the performance of wake steering applied at a section of a commercial wind farm. It is the second phase of the study in which the first phase was reported in Fleming et al. (2019). The authors implemented wake steering on two turbine pairs, and compared results with the latest FLORIS (FLOw Redirection and Induction in Steady State) model of wake steering, showing good agreement in overall energy increase. Further, although not the original intention of the study, we also used results to detect the secondary steering phenomena. Results show an overall reduction in wake losses of approximately 6.6% for the regions of operation, which corresponds to achieving roughly half of the static optimal result.

## 1 Introduction

Fleming et al. (2019) described a new field campaign at a commercial wind farm evaluating wake steering. It also presented initial results from the first phase of the campaign. Following the completion of that first phase and the subsequent article describing it, researchers made improvements to the engineering models, control design, and analysis methods. The Phase 2 campaign began in January 2019. This paper reports on the combined results of both phases.

To avoid repeating many points already made in that paper, we limit the introduction and background on wake steering in general and in theory to only what has been updated since that paper was first written. However, we will review the site and test setup in this article.

## 2 Literature Update

Since the publication of part 1, a number of papers on wake steering have been added to the literature. These papers add updates to the engineering models, control design, and field validation of wake steering and we summarize them here to add context to this quickly advancing line of research.



## 2.1 Engineering models

Engineering models provide the tools to design and analyze wind farm controllers. In this research, we rely on the FLOw Redirection and Induction in Steady State (FLORIS) tool, which includes several selectable wake models, as well as the wind farm control design and analysis tools themselves (NREL, 2019).

Important recent advances in engineering modeling of wake steering come from work that enables a more accurate description of aerodynamic effects of wake steering for turbine arrays of more than two turbines. In Fleming et al. (2018a), it was shown that counter-rotating vortices are expected to generate effects not captured in the version of FLORIS used in Phase 1 of this study, which employed the models of Bastankhah and Porté-Agel (2014); Niayifar and Porté-Agel (2015); Bastankhah and Porté-Agel (2016) to describe wake recovery and deflection. Recent work has developed models of wake steering that include

counter-rotating vortices that produce the aerodynamic effects that are the main drivers of wake steering (Martínez-Tossas et al., 2019).

    In Bastankhah and Porté-Agel (2019), a detailed wind-tunnel-based study showed that for arrays of turbines performing wake steering, the best strategy is for each successive turbine in a column to have a reduced yaw offset from the one directly upstream. Recently, the Gauss-curl hybrid (GCH) model was introduced in King et al. (2019). This model proposes an analytic

implementation of the vortices of the curl model of Martínez-Tossas et al. (2019) to modify an underlying Gauss model of Bastankhah and Porté-Agel (2014); Niayifar and Porté-Agel (2015); Bastankhah and Porté-Agel (2016). This latest model will be used in this study and a brief overview of its theory will be included in this paper; see King et al. (2019) for a full description.

## 2.2 Field validation

Since the first paper, an additional publication documenting a trial of wake steering at a commercial wind farm was published.

Howland et al. (2019) implemented a wake steering controller on an array of six turbines at a commercial wind farm and observed gains in power production for the waked cases tested.

## 2.3 Controller design

In the initial paper, we discussed that the controller, based on a lookup table of statically optimal yaw offsets that does not account for dynamic wind direction variation or the limits of the yaw controller, was likely underperforming compared to a

controller designed to account for dynamic conditions. Several recent papers propose more dynamically optimal approaches to wake steering. For example, Bossanyi (2018) introduces a new dynamic model of wakes and wind farm controls that can be used to assess the dynamic performance of wind farm controllers. Kanev (2019) proposes an elegant implementation of lookup-based, yaw-offset wake steering, which includes hysteresis, to show that, for the case study in the paper, a well-designed dynamic controller might achieve up to 67 - 75% of the static optimal.

Annoni et al. (2019) presents a method of wind direction estimation using a consensus algorithm to combine the individual turbine measurements of wind direction into an overall flow field. Combining this method with wake steering would likely



improve the accuracy and timeliness of offsets chosen by wind direction as the so-called consensus wind direction will more effectively estimate wake direction of travel versus a single point measurement from the turbine's nacelle.

Finally, Simley et al. (2019a) proposes new techniques for evaluating the statistical variation of wind direction in terms of the effect on wake propagation direction, and uses this analysis to design new dynamically optimal controllers. This work has
been used in the current, second phase of the study.

## 3   Field Campaign Overview and Update

The field campaign is located within a subsection of a larger wind farm. The subsection was chosen because of the occurrence of wake losses in common wind directions. The subsection is shown in Fig. 1.

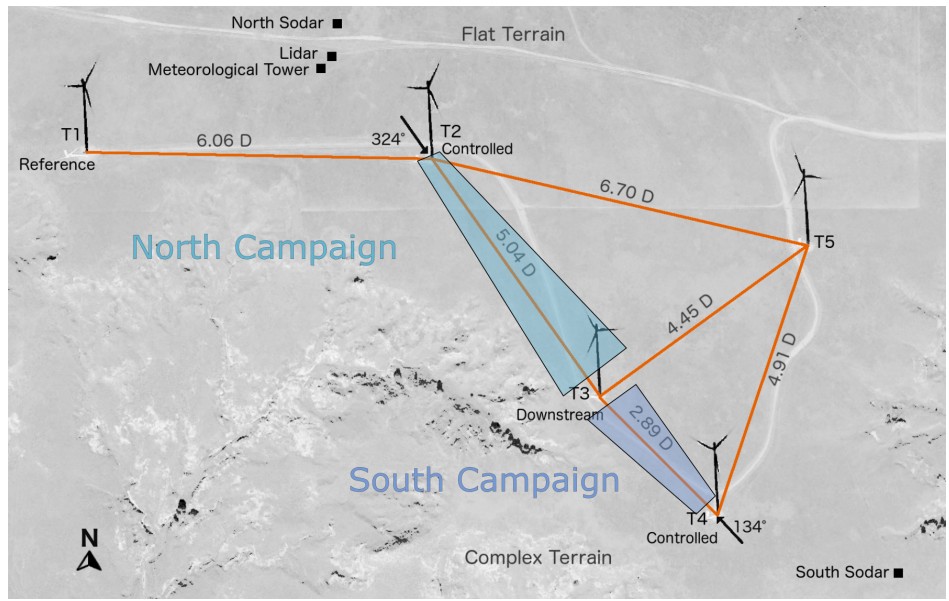

**Figure 1.** Layout of the experimental site. Turbine 2 (T2) and Turbine 4 (T4) have wake steering implemented to benefit Turbine 3 (T3), whereas Turbine 1 (T1) and Turbine 5 (T5) are reference turbines. The position of the installed meteorological equipment is also shown. Finally, the complexity of the terrain to the south and flat terrain to the north are indicated. Satellite image from © Google Earth, 2019.

As discussed in part 1, and shown in Fig. 1, the subsection includes five turbines. T2 and T4 are the controlled turbines,
and implement wake steering. T3 is the downstream turbine in the experiment. T1 and T5 are unaffected turbines used as references. Further, several measurement devices are added to the site including a profiling lidar, meteorological tower, and two sodars (a north and south), whose locations are shown in Fig. 1.

Fig. 1 also indicates the names given to the two campaigns. When the winds are from the north, such that T2 is yawed for the benefit of T3, this is referred to as the North Campaign. Similarly, in the case of south winds, T4 is yawed for the benefit
of T3 (South Campaign).





The campaign can also be divided into two phases. Phase 1 was the focus of Fleming et al. (2019), and was conducted primarily in the summer of 2018. In that period, because of the seasonal variation of winds at this location, the winds were primarily from the south, and only the South Campaign was reported in that paper.

Following the completion of the first campaign, the yaw offset schedules were updated for both controlled turbines. This
5  update will be explained in greater detail in Section 4. Following the completion of the update, testing resumed in January 2019, and data for both north and south were collected. We note that a sensor calibration issue identified toward the end of Phase 1 indicated that the smaller amount of data collected from the North Campaign in Phase 1 should not be used. This was corrected before the start of Phase 2. Total data accumulation for the two phases and both campaigns is shown in Fig. 2.

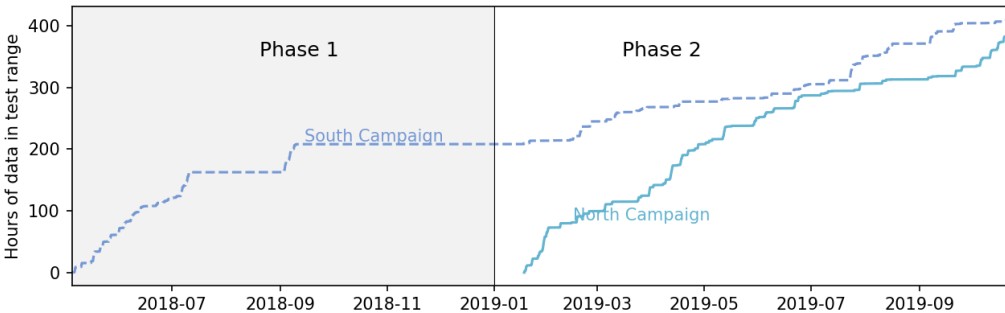

**Figure 2.** Data accumulation for both phases and campaigns. For both north and south, we consider only data that are in the range of wind directions where the control turbine would be activated, and the data are usable in that they contain no faults or issues with any sensor used in control or analysis.



## 4   Controller

In this section, we review the wake-steering controller used in this study. Fig. 3 provides the controller, as presented in the part 1 paper. The controller computes an offset vane signal to send to the (unmodified) turbine yaw controller—illustrated in the figure—which is based on analysis of the first phase (Fleming et al. (2019)) and more detailed analysis in Simley et al. (2019a).

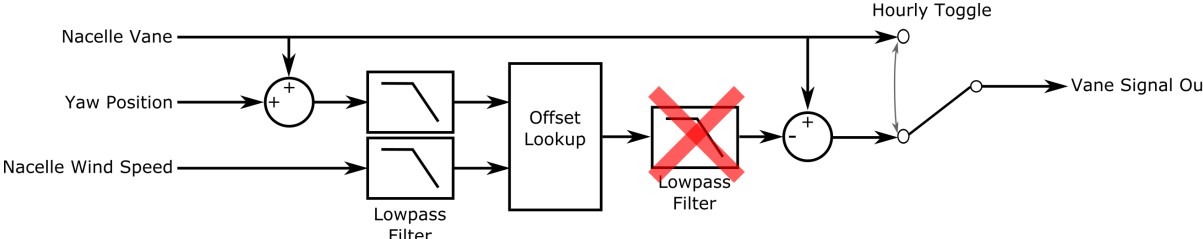

**Figure 3.** Wake steering controller used in the study. As in Phase 1, the controller produces an offset wind vane signal to be passed to the yaw controller. In Phase 2, the filter at the lookup table output was removed.

First, the filters on the input to the lookup table were adjusted, such that the filter on wind direction has a time constant of 30 s, whereas the filter on wind speed has a constant of 60 s. The lookup-table output filter was removed, as shown in part 1, because it introduced unnecessary lag to the response. Note also that the controller is toggled on and off hourly to provide comparable data sets between when the controller is on and off.

The other important change to the controllers was the offset in the lookup tables themselves. In contrast to Phase 1, in which the static optimal settings were directly deployed, the lookup table values are computed in Phase 2 by using a blend of static and robust optimization that accounts for some uncertainty in wind direction. Simley et al. (2019a) and Simley et al. (2019b) document this design process in greater detail. Finally, based on analysis from Phase 1, offsets were applied in higher wind speeds where allowed by the envelope of safe operation.

It is most likely suboptimal to implement wake control by manipulating the existing yaw controller through its vane input rather than directly modifying it; however, this was not an option for this work. Research conducted, such as by Kanev (2019), indicates that future studies using carefully designed direct modifications to the yaw controllers can improve on this work.

### 4.1   Controller Performance

This section reviews the performance of the controller in achieving the desired offset behavior. Fig. 4 reviews the conventions to be used throughout this article. A positive yaw offset is defined to be counterclockwise. When the controller is toggled off, no offset is applied (called "baseline"), whereas when the controller is on, an offset is applied (called "controlled"). In all figures, blue is associated with the baseline and magenta represents controlled. Note the colors are changed from Fleming et al. (2019) in order to improve color-blindness accessibility. The hourly toggling in the controller is performed to provide approximately equivalent distributions of wind speeds and directions for the baseline and controlled data sets.





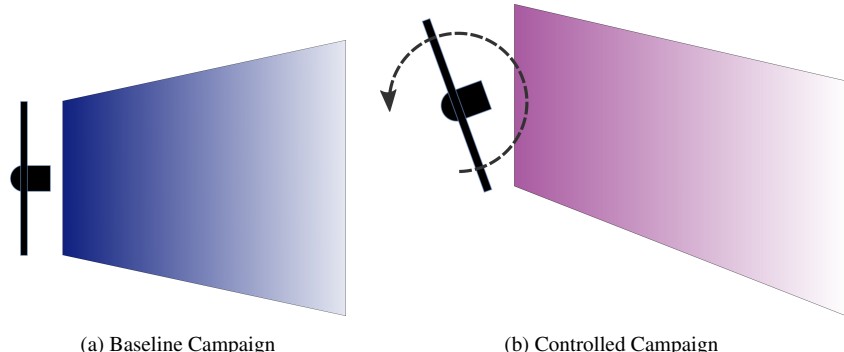

|(a) Baseline Campaign|(b) Controlled Campaign|

**Figure 4.** Baseline results (shown in blue throughout this article) are those in which the control turbine operates normally, based on the toggle setting in Fig. 3. Controlled operation includes all times when the controller is enabled according to the toggle (regardless of achieved offset); shown in magenta throughout this article. Finally, the figure indicates a positive yaw offset that represents a counterclockwise rotation of the turbine viewed from above.

Fig. 5 summarizes all of the yaw offset data by wind speed and direction for both campaigns and both phases observed over the course of the campaign. The targeted offset is shown in black, whereas the achieved offset is shown in magenta. Note the achieved offset is calculated with respect to the reference wind direction (not the wind direction measured by the turbine itself as this could be affected by the yawing). The reference wind direction for the South Campaign described in the part 1 study was provided by the south sodar. However, in the present work, we found a better choice: an average of measurements. For the North Campaign, this is the average of the wind direction measurements made by the lidar, as well as T1 and T5's wind direction measurement computed using the nacelle vane and measurement of yaw heading. For the South Campaign, this is T1, T5, and the south sodar averaged.

Shaded regions are overlayed in Fig. 5 to denote certain wind direction areas. The total area of the shaded region (orange and blue) indicates the range of wind directions for which a yaw offset is observed and will be used in later plots to indicate where control is and is not applied in analysis figures. The region is subdivided into a light blue region that indicates that yaw offset is applied and desired, whereas the orange region indicates that the yaw offset is applied unintentionally. Given the constraints of the yaw control system, such a region is probably somewhat inevitable if larger offsets are to be achieved at all. However, as described earlier, optimal wake steering control is an area of active research, and analysis of the controllers ability to achieve the desired offset in the face of uncertain wind direction variations is a subject of ongoing work (Simley et al. (2019a); Kanev (2019); Rott et al. (2018); Quick et al. (2017); Bossanyi (2018).

A final analysis considers the success of hourly toggling in balancing (between the baseline and controlled) the inflow conditions observed. This is shown in Fig. 6, in terms of histograms of wind speeds and directions observed for the North and South campaigns. Dividing the data into the baseline and controlled data sets shows that the two conditions are fairly well-balanced with both settings seeing similar overall conditions. In addition, one can observe that the North Campaign experienced faster overall conditions, which is expected for this location.

**Figure 5.** Comparison of targeted versus achieved offsets for the North and South campaigns. The black lines indicate the targeted offset for a given wind speed and direction. The magenta lines show the average achieved offset (with 1-minute average points shown to indicate the spread in values going into that average). Colored bands are used to indicate where a yaw offset is achieved intentionally (light blue) and unintentionally (orange). The combined orange-blue region is the range of directions wherein the controller is active. The South Campaign, as discussed earlier, includes two phases of control; however, this figure indicates that the achieved offsets are similar).

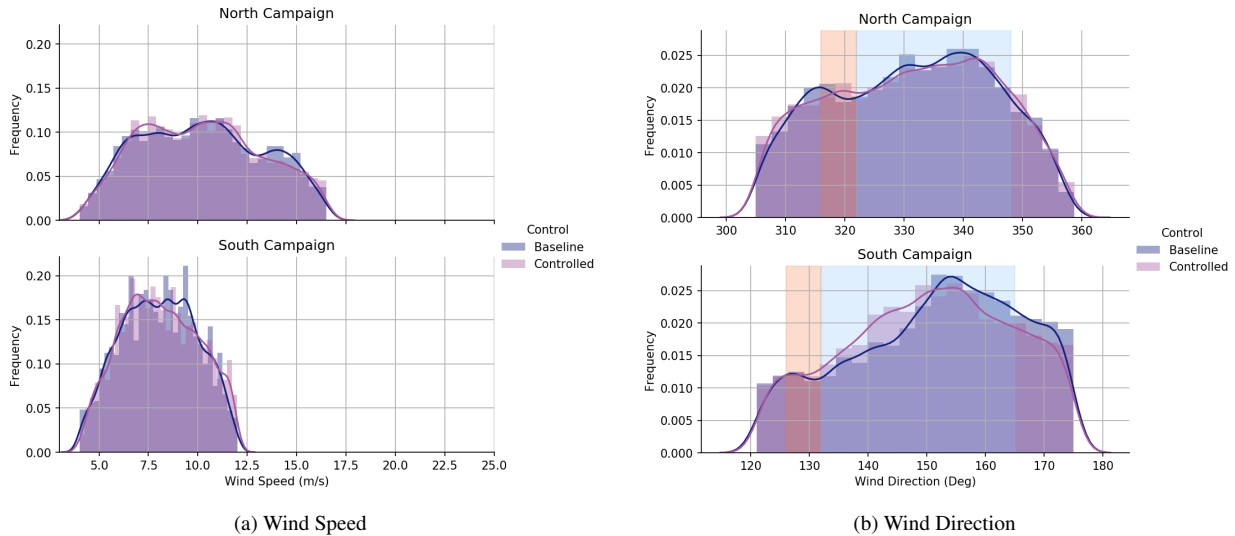

(a) Wind Speed

(b) Wind Direction

**Figure 6.** Comparison of wind speed and wind direction histograms, which is divided into baseline and controlled operation. Note the bands of desired and nondesired yaw offsets first shown in Fig. 5 are used here and throughout the article to illustrate the controlled region.



| Parameter | Value |
| --- | --- |
| ka | 0.38 |
| kb | 0.004 |
| alpha | 0.58 |
| beta | 0.077 |
| ad | 0 |
| bd | 0 |
| Turbulence Intensity North | 0.1 |
| Turbulence Intensity South | 0.08 |

**Table 1.** FLORIS Gauss model parameterization

## 5   FLORIS

FLORIS is wind farm control software tool, which includes wake models, as well as wind farm control design and analysis tools (NREL, 2019). It is central to this work as it was used to design the test controls for this study, and validating its ability to predict gains from wind farm control is a key outcome. FLORIS was co-developed by NREL and the Delft University of Technology.

FLORIS includes several optional selections for the wake model, including the original multizone model (see Gebraad et al. (2016)), the new vortex-based curl model discussed in Martínez-Tossas et al. (2019), and the Gaussian wake model of Bastankhah and Porté-Agel (2014); Niayifar and Porté-Agel (2015); Bastankhah and Porté-Agel (2016).

In part 1, the Gauss model was selected for the design and analysis of the experiment as it is the current standard at NREL and often used in literature. The model includes several tunable parameters. In this work, we use the "default" settings for the wake deficit model provided in Niayifar and Porté-Agel (2015); Bastankhah and Porté-Agel (2016) and tune only turbulence intensity (TI) separately for the North and South campaigns.

In part 1, we allowed TI to be set for each observation based on the south sodar measurement, but in the present work, we select a best-fit overall TI. Specifically, the TI is selected to provide a close match between the baseline wake losses in FLORIS to those measured in the field. The selected parameters are summarized in Table 1.

It is important to note that the chosen TI values for the model are likely aggregating multiple atmospheric effects. For example, the present version of FLORIS lacks a true near-wake model, and so this may explain the lower TI for the South Campaign. FLORIS currently underestimates near-wake losses, so this is currently corrected by a lower average TI than is physically occurring. Improving near-wake models is a focus of future work.

For the wake deflection model, we initially selected, in part 1, the deflection model described in Bastankhah and Porté-Agel (2014); Niayifar and Porté-Agel (2015); Bastankhah and Porté-Agel (2016). The default model directly provides no free parameter to tune the gain from wake steering, as is done in the model from Jiménez et al. (2010). In previous work, a multiplication on the initial deflection angle as a result of wake steering was added to allow for a better match to the gain





in power seen, for example, in Simulator fOr Wind Farm Application (SOWFA) simulations. Thus, the Gauss model with a multiplication on deflection was employed to design the control strategy used in Phase 2, along with the uncertain optimization of Simley et al. (2019a). A deflection multiplier of 2 provides a better fit to the 5D-spaced North Campaign, whereas a multiplier of 1 fit better with the 3D-spaced South Campaign. This inability to match both ranges with a single parameter is likely because

of the fact that this gain on deflection does not capture the underlying physical mechanisms of wakes affected by wake steering. The changes to the wake, which include the generation of counter-rotating vortices, are described more accurately in the curl model of Martínez-Tossas et al. (2019).

More recently, the GCH model described earlier, which starts at the base with the Gauss model of Bastankhah and Porté-Agel (2014); Niayifar and Porté-Agel (2015); Bastankhah and Porté-Agel (2016) but is then modified analytically by equations

adapted from Martínez-Tossas et al. (2019), was introduced in King et al. (2019). This model combines the advantages of fast computation and tunability of the underlying Gauss model with the included physics of counter-rotating vortices responsible for secondary steering (see Fleming et al. (2018b)) and yaw-added wake recovery, which models the additional gains in power as an additional increase in flow velocity driven by the counter-rotating vortices. For a detailed description, see King et al. (2019).

For the remainder of this article, we will refer to three wake models within FLORIS (Annoni et al., 2016). Gauss is the implementation of Bastankhah and Porté-Agel (2014); Niayifar and Porté-Agel (2015); Bastankhah and Porté-Agel (2016), whereas "Gauss-2x" includes the gain on deflection and finally the new GCH model. Although the primary purpose of the GCH model is to improve secondary steering predictions, we also show that it can provide the needed increase in energy gain formerly produced by multiplying the deflection amount. Therefore, the plots will often include results for Gauss, Gauss-2x,

and GCH.



## 6   Analysis

Having described the controller, field campaign, and engineering model FLORIS, the remainder of this article will focus on comparing the collected results, with predictions, or, more accurately, resimulations of the results in FLORIS.

In this article, we use the same method for comparing energy production, the "balanced energy ratio" method, described in the previous paper (Fleming et al., 2019). The only update is that in that paper, a single turbine provides the reference power, while a single sensor provides the wind direction. Experimentation showed that a more precise result was obtained when the direction was set as the average wind direction of the two reference turbines (T1 and T5), with the wind direction measured by the lidar for the north and sodar for the south (using a weighted-average over the rotor heights for the lidar and sodar). Similarly, the reference power is the average of T1 and T5 instead of just T1 for the North Campaign; however, it is only T1 for the South Campaign, as including T5 into the analysis in this case made the baseline energy ratio noisier, presumably because the wind from the south arrives at T5 over complex terrain.

For the FLORIS resimualtions, the wind speeds, directions, and yaw offsets measured are applied to the various models (Gauss, Gauss-2x, and GCH), and a predicted power of the reference and test turbines are produced, then the analysis is identically performed as computed for the field data. The Gauss, Gauss-2x, and GCH models are simulated using the measured offsets (comparing the nacelle position of the control turbine with the wind direction). A final FLORIS simulation is performed, again with GCH; however, using the targeted, rather than the achieved, yaw offsets. This simulation will be referred to as GCH optimal, as it is meant to represent the result if the dynamic controller achieved the static targets exactly.

A final note is that the Python implementation of the balanced energy ratio method of comparison is included in the FLORIS repository.



## 7 Results

The results from both the North and South campaigns are presented in this section using the methods explained in Section 6. This analysis recovers Phase 1 data analyzed in part 1, and combines with Phase 2, thereby replacing the results with the full data set.

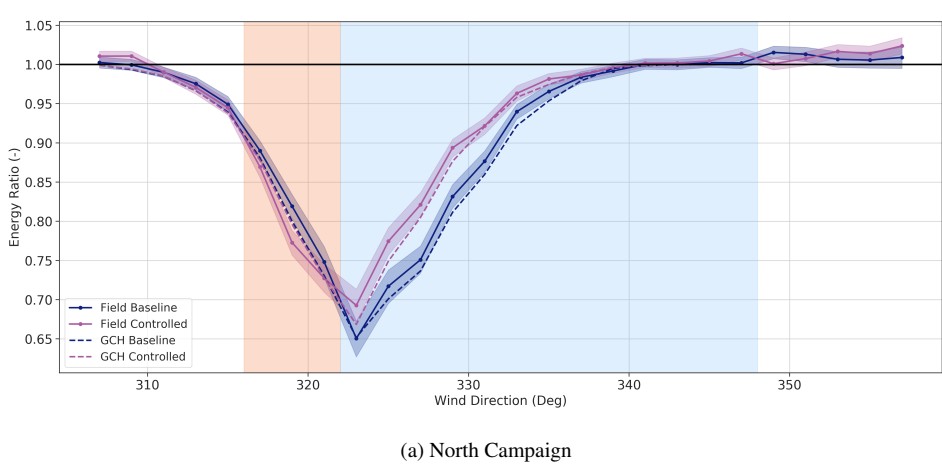

(a) North Campaign

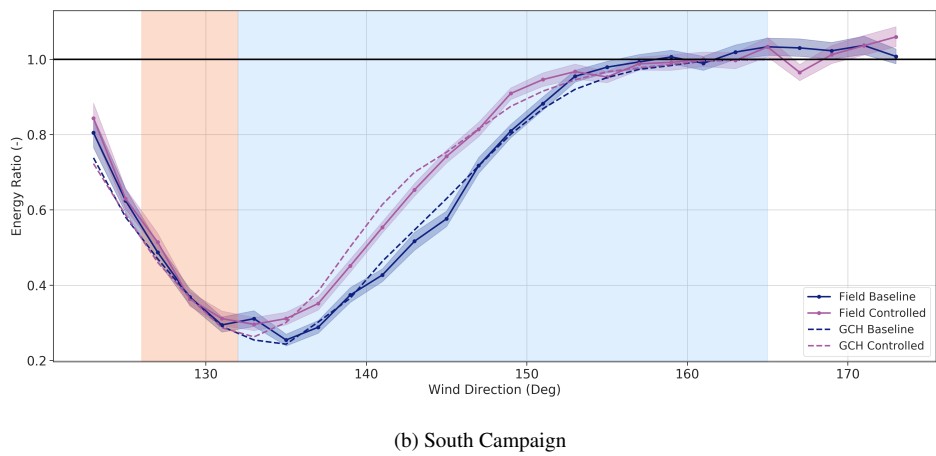

(b) South Campaign

**Figure 7.** Energy ratio of T3 for the North and South campaigns. For both campaigns, this represents the ratio of energy produced by T3 with respect to unwaked reference turbines. The banded region, as explained in Fig. 5, indicates regions of offset activity, either intended (light blue) or unintended (orange).

5      The energy ratios for the downstream turbine (T3) are provided for both the North and South campaigns in Fig. 7, whereas the difference between the baseline and controlled energy ratios is shown in Fig. 8.

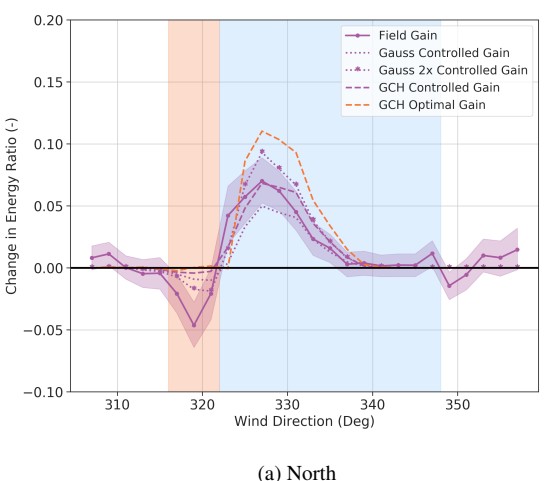

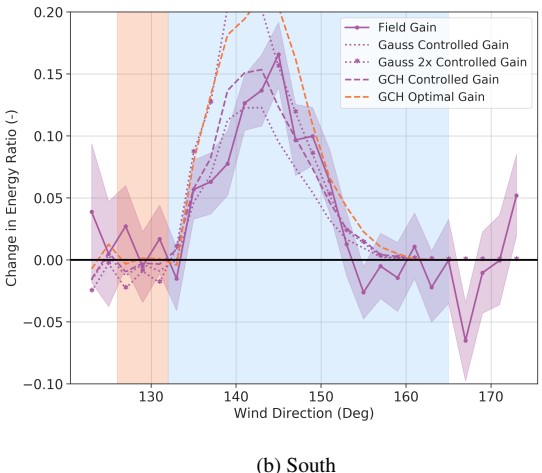

(a) North

(b) South

**Figure 8.** Change in energy ratio for the downstream turbine (T3) with respect to the reference. FLORIS results include "controlled" gain, which simulates the results in FLORIS using the measured offsets from the field campaign for each model, whereas the "optimal" gain is computed using the targeted offsets, for GCH only.

The results show a clear gain in energy production for the downstream turbine for both the South and North campaigns. The peak gains are underforecasted by the Gauss model in both cases and overforecasted in the Gauss-2x model. The GCH model somewhat underpredicts the peak gains but is in general a reasonable fit.

The optimal gain (the expected gain if desired offset is always achieved) is higher than the realized gain. As stated, this field campaign uses a first-pass lookup-table method to offset the vane signal provided to the usual yaw controller. We believe the optimal performance can be more nearly achieved by direct control over yaw (as in Kanev (2019)), accounting for yaw control limitations in design (Simley et al. (2019a)), and by improving measurement of wind direction (for instance, using information sharing between turbines (Annoni et al. (2019))). However, realizing perfectly the optimal results is not possible as this would imply excessive yawing and perfect information of wind direction.

One point of ambiguity in the results shown in Fig. 7 and Fig. 8 is the effect of "wrong-way steering," (i.e., the region of unintentional wake steering in the red-banded region). The loss in this region is more than expected in the North Campaign, and less than expected in the South Campaign. Comparing the models shows that GCH expects less losses than the previous models. However, this is likely related to an issue in modeling near wakes, which both cases represent (3D and 5D spacing) to some extent. In King et al. (2019), GCH predicts wrong-way steering better at distances above 5D.

An interesting additional insight comes from dividing the data in Fig. 8a into daytime and nighttime conditions. This is shown in Fig. 9. This approach reveals a pattern of lower gains in the daytime and higher gains in the nighttime. This makes sense, given stable, low-TI conditions at night producing deeper wake losses, and less meandering of the wakes. It is a useful reminder that the overall results of Fig. 8a are the average of two somewhat different conditions. FLORIS currently models the average,





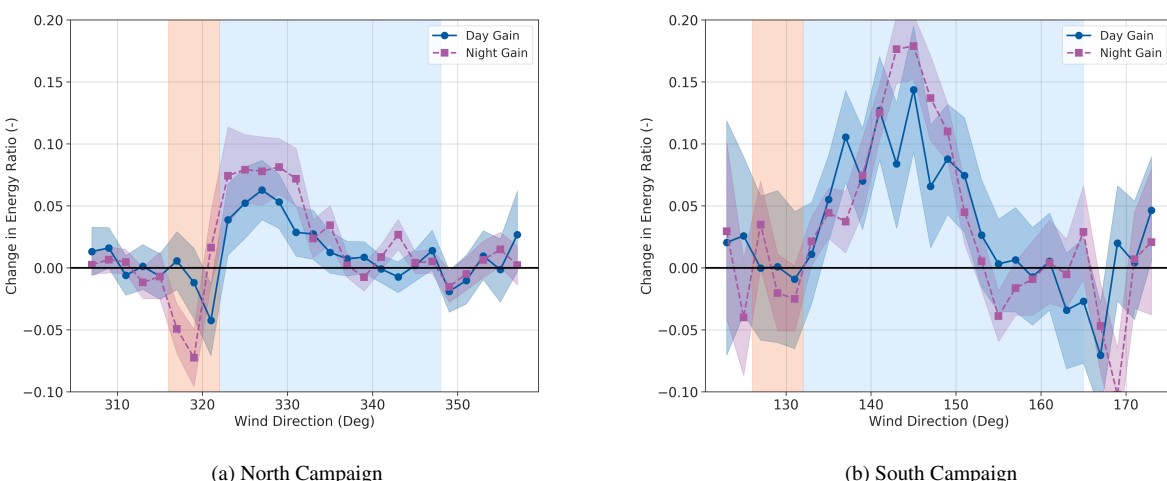

| (a) North Campaign | (b) South Campaign |

**Figure 9.** Comparison of change in energy ratio of T3 for daytime versus nighttime conditions.

but a division into two separate conditions, as in what is proposed in Ruisi and Bossanyi (2019), or as an additional continually varying signal input to FLORIS, which might enable more tailored control settings and improve results further. A future paper will in greater detail analyze the differences in performance relative to measured turbulence and stability characteristics. The turbulence and stability characteristics of the site for the south campaign are presented in Murphy et al. (2019).

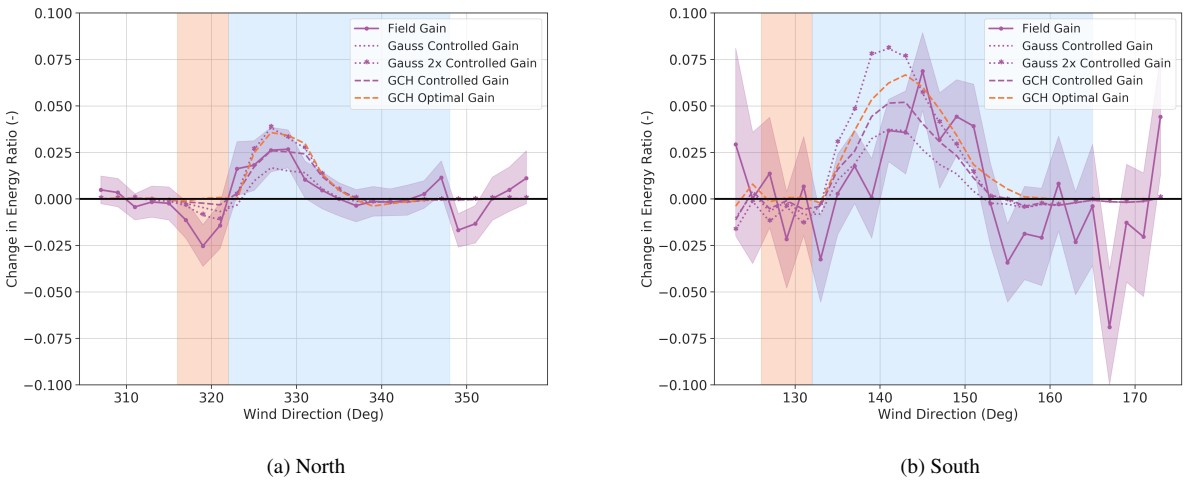

| (a) North | (b) South |

**Figure 10.** Combined change in energy ratio, which is the total energy production of T2 and T3 for the North Campaign, and T4 and T3 for the South Campaign.





The change in total energy production for the upstream and downstream turbines combined is shown in Fig. 10. This figure includes the losses from the upstream turbine with the gains downstream. FLORIS currently models losses as a result of yawing via the pP exponent, which is the exponent of the cosine of the yaw angle (ie, if pP equals 2, power is lost as a function of the cosine squared of the yaw angle.) In this work pP is set to 1.9. The results again show a good match between the field

5   results and the GCH model in FLORIS in the main wind directions for improvement, but some losses on the outer regions. One possibility in the South Campaign is that the wind direction estimate for T4 is slightly off, which is possible because there are fewer measurements to compare to in the south, and the terrain is much more complex, and, finally, there is more measurement data available for the north, including the lidar. Although it does not show as an offset in Fig. 5, it would help to explain the lack of loss in the unintended region, the shift of the peak to the right, and the unnecessary losses above 152 degrees in Fig. 10. It

10   may also be that the flow from the south, which passes over very complex terrain, is less homogeneous and therefore interacts with the wakes in more complicated ways.



## 7.1 Overall results

To assess the overall effect of wake steering, we define the wake loss as the total difference in energy production by the waked turbine (or combined energy production of waked and controlled turbine) versus the reference turbine or turbines. To control for the impact of differences in wind distributions between the baseline and controlled sets, we first compute the average wake

5   loss per wind speed and wind direction bin, and then compute a weighted sum across these bins, where the weight is the total number of points (baseline and controlled) in each bin. The wake loss expressed as a percentage is then the ratio of this total amount over the total amount of energy produced by the reference turbine or turbines.

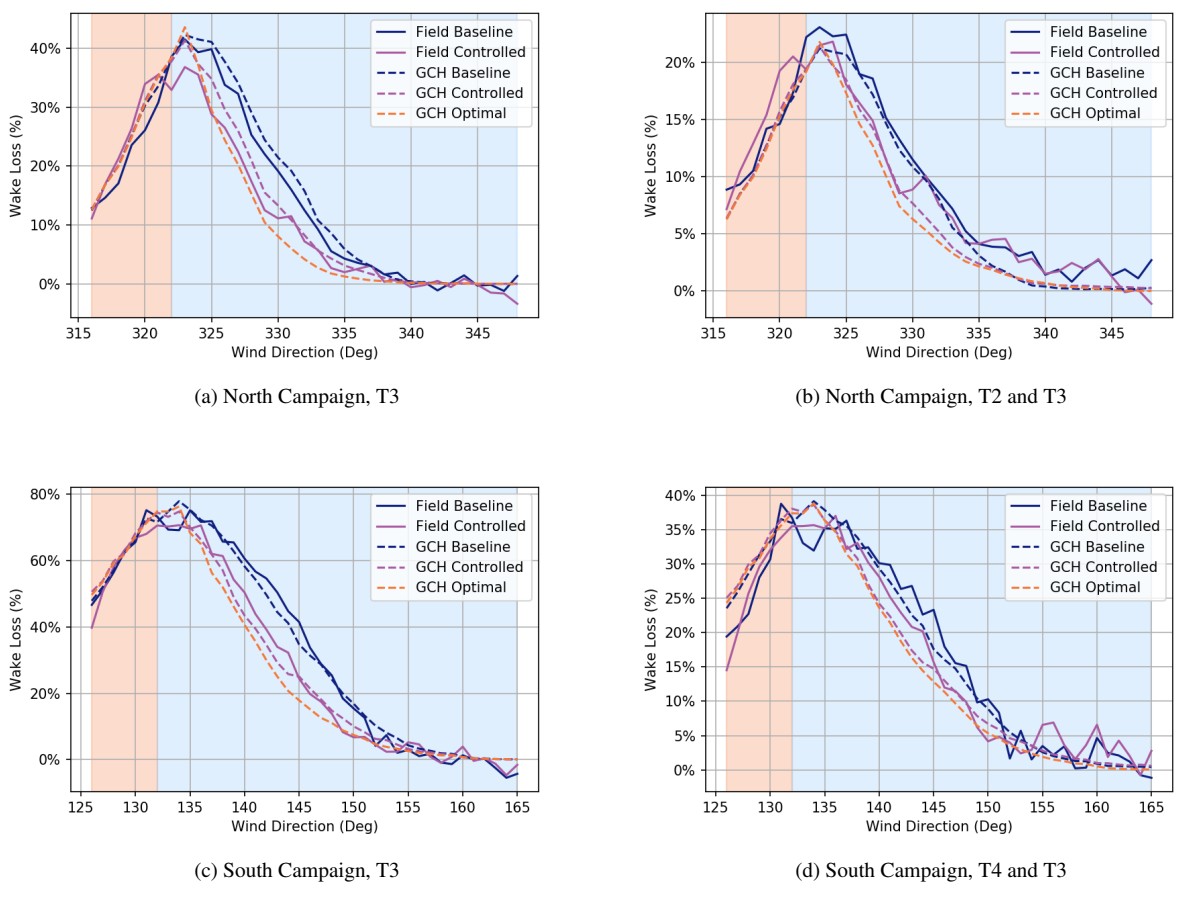

**Figure 11.** Wake losses computed per wind direction bin for the North and South campaigns.

Computing this result per wind direction bin produces the plots shown in Fig 11. These plots confirm that the wake loss calculation yields a similar result in plots against wind direction as the energy ratio plots shown earlier. It is now possible to

10   compute the overall effect on reducing wake losses for both campaigns, which is the same calculation computed as in Fig 11, but now across all wind directions as well, thereby yielding a total overall value. These results are summarized in Table 2.





|  | North | | South | |
|---|---|---|---|---|
|  | T3 | T2 and T3 | T3 | T4 and T3 |
| Field | 14.4% | 6.6% | 13.4% | 6.4% |
| Gauss | 9.1% | 2.9% | 9.1% | 3.5% |
| Gauss 2x | 17.3% | 11.0% | 16.2% | 10.4% |
| GCH Controlled | 15.0% | 9.0% | 12.1% | 6.6% |
| GCH Optimal | 24.4% | 12.7% | 18.2% | 10.4% |

**Table 2.** Overall percent reduction in wake losses across the controlled region

Table 2 shows that the overall reduction for the combined turbines of the North and South campaigns is 6.6% and 6.4%, respectively. This corresponds to between half and 60% of the static optimal gain predicted by GCH. Compared to the model outputs using the actual achieved yaw angles, the results are fairly close to GCH, while being significantly above Gauss and significantly below Gauss 2x.





## 8 Vortex behaviors and secondary steering

An additional output of this study is to use the three turbines in a row (T2, T3, and T4) during the wake control to study secondary steering. Recent research has focused on the important role counter-rotating vortices created in wake steering (Howland et al. (2016)) play in determining the behavior of the steered wake (Vollmer et al. (2016)), especially for arrays of turbines larger than two (Fleming et al. (2018a); Martínez-Tossas et al. (2019); King et al. (2019)). Although not the focus of the study at the beginning, the field data have been analyzed to assess the presence of the effects in the field data.

One important prediction from this research into vortex-based explanations of wake steering is a phenomena called secondary steering Fleming et al. (2018a). Secondary steering, as observed in large-eddy simulations of wake steering, for example, shows that the wake of a nonsteered turbine, if it is itself in the wake of a turbine performing wake steering (and thereby generating the counter-rotating vortices that propagate downstream), will in fact be steered. This means that the version for FLORIS using the Gaussian wake model of Bastankhah and Porté-Agel (2014) will underpredict the change in energy on a third turbine in a row, because it will only account for the change in the wake of the first turbine, and not the change owed to secondary steering. The GCH model, on the other hand, will include this effect.

Fig. 12 shows the energy ratios for the third turbine in the row for the two campaigns while the change is shown in Fig. 13.

The first observation is that based on Fig.12 and Fig. 13) the impact on the third turbine is clearly observed. For the North Campaign, GCH improves the estimate of the increase in power of the third turbine (T4) by including secondary steering. The results are less clear for the South Campaign. The shift in the field-measured nadir of baseline power to the right implies that secondary steering would not increase power until more northerly wind directions than expected, but that only explains part of the discrepancy. The very close spacing of T4 and T3, or the complex terrain in the inflow to the South Campaign, could also be part of the explanation.

Finally, in a presentation at the Wind Energy Science Conference 2019, Safak Altun showed that wake steering can produce a change in the wind direction downstream from the turbine implementing the steering (Altun, 2019). In Fig. 14, we compare T3's alignment with respect to the reference wind direction for the baseline and controlled sets and observed that it does appear to offset itself for the areas with highest offsets for T2, suggesting that T3 is observing a change in wind direction as a result of wake steering.



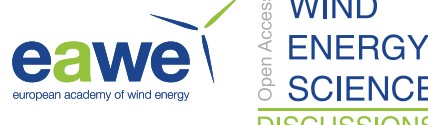

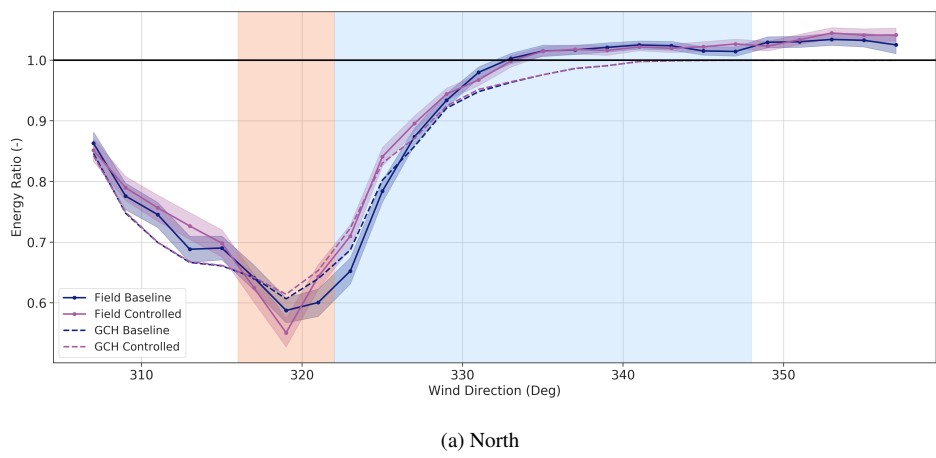

(a) North

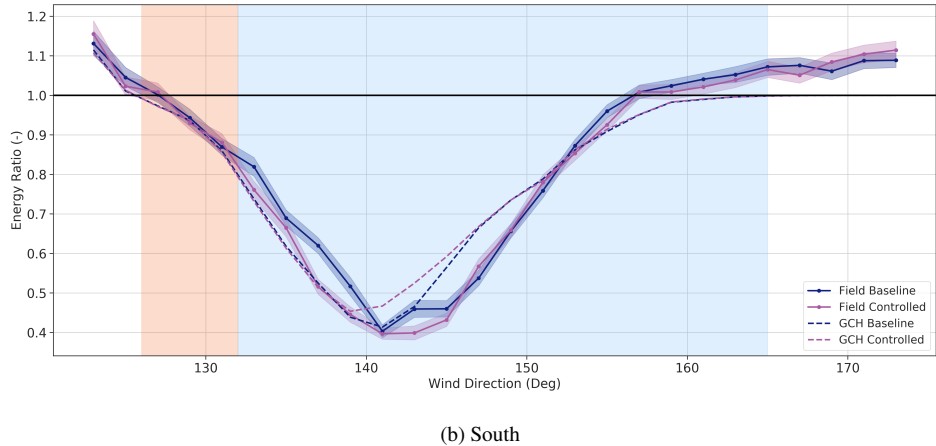

(b) South

**Figure 12.** Energy ratios of the third turbine in the row for the North Campaign (T4) and the South Campaign (T2). The effect of secondary steering is helpful on the right side of the nadir, and harmful otherwise. The mismatch in the baseline nadir of the South Campaign suggests directional calibration issues.

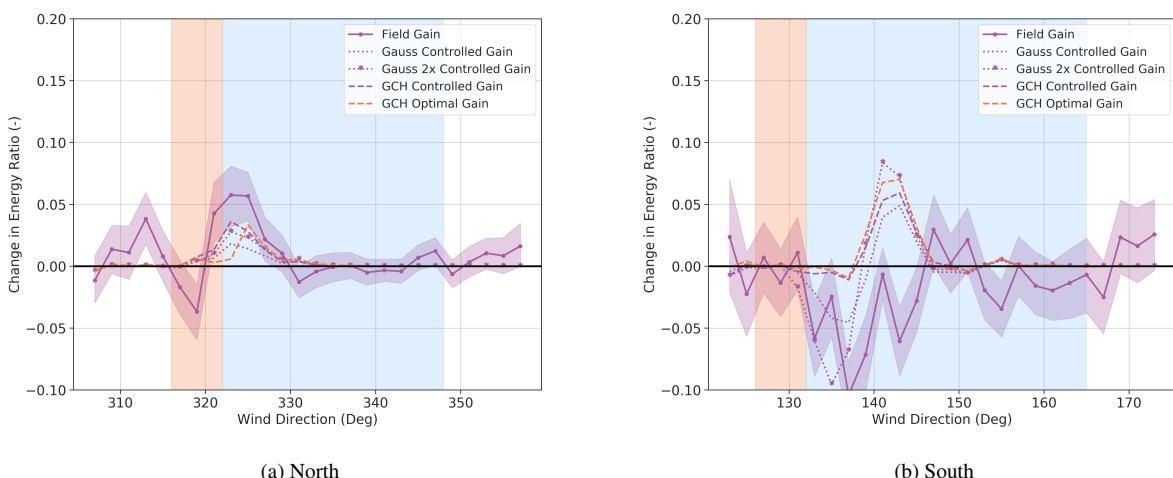

**Figure 13.** Change in energy ratio of the third turbine in the row for the North Campaign (T4) and the South Campaign (T2).

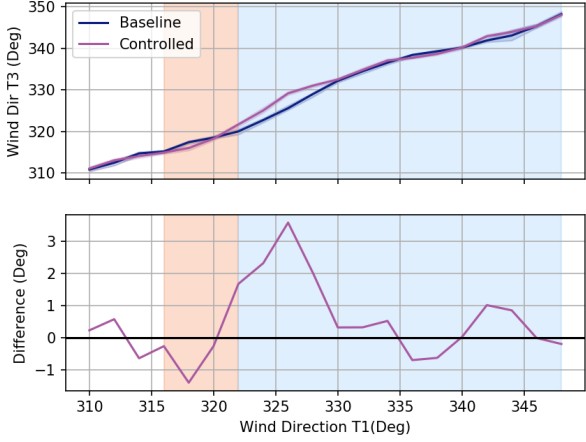

**Figure 14.** T3's alignment with the reference wind direction in baseline and controlled conditions indicates an apparent change of wind direction for wind directions with maximal offset.





## 9    Discussion and Conclusions

This article reports the results of a 16-month field campaign assessing the ability of wake steering to increase the energy production of turbines within a wind farm. The results show that, for both the North and South campaigns, the energy is increased for the two-turbine pairs when steering is applied. Further, the gains in the areas of largest applied offsets match very well to the predictions of the engineering model FLORIS.

Additionally, the presence of a third turbine in both campaigns allowed for a demonstration of the secondary steering effect in a commercial farm. We expect this effect to be critical for the design of wind farm controls for large arrays. King et al. (2019) show that models without secondary steering increasingly underpredict power impacts as the number of turbines in an array increases. Studies such as Bastankhah and Porté-Agel (2019) demonstrate the important influence these effects will have on the design of the optimal controller, as accounting for the interaction of steered wakes leads to different optimal angles than assumptions of independence would imply.

Overall, for the North and South campaigns, we report a reduction in wake losses of 6.6% and 6.4%, which is roughly half of the static optimal values predicted by the GCH/FLORIS models. At a high level, for the wind directions most studied by large-eddy simulation—the regions of largest gain—performance was nearly optimal, whereas in the outer regions there was underperformance. There was too much loss in the "unintended yawing" region for the North Campaign, and too much unproductive yaw activity in the partial wake region (153–165 degrees) of the South Campaign.

We find this to be a very exciting result, as we believe that there are still more opportunities for improved performance for the next generation of wind farm controllers to approach higher percentages of the static optimum. This result represents the gain in energy produced using a precomputed lookup table to implement an offset control strategy using the turbine's measurement of wind speed and direction. We applied the offset by offsetting the vane signal provided to the unmodified yaw control system.

There is an opportunity for continued research into the robust optimal lookup table (as opposed to the static optimal). Simley et al. (2019a); Rott et al. (2018); Quick et al. (2017) present opportunities to design lookup tables that perform optimally for specific atmospheric characteristics and turbine yaw control design given uncertainty assumptions in the wind and controls.

Further, better performance is likely when the yaw offset control can be implemented directly, rather than by manipulating the vane input of the existing yaw controller. Designs such as those presented in Kanev (2019) could then be implemented. The underprediction in achieving desired yaw angles in Fig. 5 is likely a consequence of having only indirect control over yawing. In general, the ability to achieve larger offsets in desired regions while avoiding them in undesired regions can only be improved through greater direct control.

Obtaining better knowledge of the inflow conditions will also improve performance. The consensus control algorithm of Annoni et al. (2019) provides a means for turbines to cooperate when estimating the wind flow in real time. This estimated consensus wind field can include spatial filtering and even preview, which could be of much use to the typically slow yaw controller. Other possibilities include incorporating the direct measurement of the inflow itself in the controls (Raach et al., 2019).





Finally, improved models of complex effects, such as vortex behaviors and curl, provide opportunities for control strategies to optimally exploit flow control and increased entrainment of energy into the wind farm. These improvements will raise the estimate of gains of wake steering (King et al., 2019).

Validation in realistic conditions is a hurdle to the broader adoption of wake steering this article is attempting to address.
However, it does not address another barrier, which is the impact on loads from wake steering. There is existing literature on the topic of how yaw misalignment impacts loads (e.g., see Kragh and Hansen (2013), Damiani et al. (2017), Schulz et al. (2017), and White et al. (2018)); however, the results are complex. Yaw misalignment impacts various turbine component loads differently (e.g., it can reduce or increase blade loads), and the effect can depend on turbine details, control settings, and conditions. Other studies seek to assess the impact of loads including the effect of dewaking downstream turbines (for
example, see Mendez Reyes et al. (2019)). Advances in the general understanding of the overall impact on turbine lifetime and maintenance needs would be helpful.

Another issue noted in this article is that the model used for the baseline in all cases is the Gauss model, and we note a tendency toward underpredicting wake losses even when assuming a rather low fixed annual turbulence intensity. A near-wake model, such as presented in Ishihara and Qian (2018) or Blondel and Cathelain (2020), could improve the fit of the closer-
spaced turbines without relying on a lower turbulence setting. For the cases of the third turbine in a row, we propose that new turbulence models or deep-array models could help increase the accuracy of wake losses in the model even assuming higher turbulence.

Finally, the methods used to assess the performance of the wind farm controllers represent an interesting opportunity to apply sophisticated statistical methods to assess the true net gain in energy over the counter-factual case, in which the wake
steering controller was not run in identical conditions. As mentioned, the balanced energy ratio method used in this work is included with the FLORIS repository.

**Copyright Statement**

This work was authored by the National Renewable Energy Laboratory, operated by Alliance for Sustainable Energy, LLC, for the U.S. Department of Energy (DOE) under Contract No. DE-AC36-08GO28308. Funding provided by the U.S. Department
of Energy Office of Energy Efficiency and Renewable Energy Wind Energy Technologies Office. The views expressed in the article do not necessarily represent the views of the DOE or the U.S. Government. The U.S. Government retains and the publisher, by accepting the article for publication, acknowledges that the U.S. Government retains a nonexclusive, paid-up, irrevocable, worldwide license to publish or reproduce the published form of this work, or allow others to do so, for U.S. Government purposes.



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
