# Peer review of "Continued Results from a Field Campaign of Wake Steering Applied at a Commercial Wind Farm: Part 2"

_Wind Energy Science, 2019_

## Referee Comment (RC1) · Anonymous Referee #1 · 21 Feb 2020

This paper reports on a field test campaign for wake steering. Such field tests are vital to increase confidence in the usefulness of wake steering, while being difficult to design, and producing results which can be difficult to analyse conclusively. This campaign appears to have been well designed, and to have run for long enough to generate enough data to allow useful conclusions to be drawn. The paper presents a very clear description of the tests, of the analysis methods used, and the results obtained from the analysis, and is a valuable addition to the literature on this topic.

Although the paper is acceptable as it is, I have three minor comments to consider in any revision:

[Figure]

Section 6, page 11, line 10: the justification for the decision not to use T5 data in the South campaign is that the results are noisier (for credible reasons). It is important in any such analysis to demonstrate every effort to avoid unconscious bias that might result from such decisions, so it would be helpful to present some evidence to justify this decision.

Same page, line 12: typo: 'resimulations'.

Section 7.1, page 16, last line: the 'overall' figures for wake loss reduction in Table 2 are described as 'across all wind directions'. Is this an average across the bins, or an average weighted by the number of points in each bin?

---

## Referee Comment (RC2) · Torben Knudsen (Referee) · 4 Mar 2020

**Review - Continued Results from a Field Campaign of Wake Steering Applied at a Commercial Wind Farm: Part 2**

Torben Knudsen, Aalborg University

March 4, 2020

**Summary**

This paper is well written with substantial contributions. Specifically, it is of value to provide full scale tests that prove gains from using wind farm control. Publication is therefore suggested after minor revision.

**Specific comments**

1. P7 Figure 5 caption. "Colored bands are used to indicate where a yaw offset is achieved intentionally (light blue) and unintentionally (orange)."

   - The notion of intentionally versus unintentionally is a bit confusing.
   - When the active yaw controller yaws the turbine it is intended other wise the controller would have been designed to do something else?
   - However, the measured combination of yaw, wind direction and speed is not at the ideal (steady state) values. This is of course not what is preferred/intended but just a consequence of actuator dynamics, limits, uncertainty etc. which are standard in any control task?
   - Why call it unintended when there is a deviation between measured yaw and ideal yaw and the ideal is zero yaw but intended when there is an error but the ideal yaw is positive?

2. P9 "Specifically, the TI is selected to provide a close match between the baseline wake losses in FLORIS to those measured in the field." Isn't it strange to use the TI to math wake losses? Shouldn't you use the field measure TI and then perhaps adjust wake parameters to match wake losses? (Now I see that the explanation is partly in the next sentence.)

3. P10 "secondary steering (see Fleming et al. (2018b))" Please explain what that is?

4. P 13 "The optimal gain (the expected gain if desired offset is always achieved) is higher than the realized gain."

   - Will there be power gain results for the combination of the controlled turbine and the one in wake? (I will probably know when I have finished the paper)
   - The problem of lower power gains compared to ideal static calculations do to the dynamics has been pointed to long back Knudsen et al. [2015]

5. Figure 7 "this represents the ratio of energy produced by T3 with respect to unwaked reference turbines."

   - If the controller were toggle on/of e.g. every 5 minutes you could compare with the same turbine in order not to risk differences between locations.

6. P21 "We find this to be a very exciting result, as we believe that there are still more opportunities for improved performance for the next generation of wind farm controllers to approach higher percentages of the static optimum."

   - As pointed out before we can not expect to reach the static optimum Knudsen et al. [2015].

7. P21 "Obtaining better knowledge of the inflow conditions will also improve performance".

   - A candidate method for this is model based state estimation as discussed in Doekemeijer et al. [2018].

**References**

B. M. Doekemeijer, S. Boersma, L. Y. Pao, T. Knudsen, and J.-W. van Wingerden. Online model calibration for a simplified les model in pursuit of real-time closed-loop wind farm control. *Wind Energy Science*, 3(2):749–765, 2018. doi: 10.5194/wes-3-749-2018. URL https://www.wind-energ-sci.net/3/749/2018/.

T. Knudsen, T. Bak, and M. Svenstrup. Survey of wind farm control - power and fatigue optimization. *Wind Energy*, 18(8):1333–1351, August 2015. doi: 10.1002/we.1760. Published online 9 May 2014 in Wiley Online Library (onlinelibrary.wiley.com).

---

## Referee Comment (RC3) · Stoyan Kanev (Referee) · 17 Mar 2020

General comments:

The paper is very well written and clear (as usual). It provides the results from the second phase of field trials with wake redirection. These results reconfirm and strengthen earlier findings (by the authors) indicating that wake redirection control has the potential to increase the power production of a wind farm. The paper provides very valuable insights into the operation of wake steering in the field and indicates possible improvement possibilities for the future.

Minor comments:

- page 2, line 19-21: please mention if the results in Howland (2019) are in line with yours, and explain any differences between their experimental setup and approach and yours.

- page 6, line 5: why is averaging the wind directions considered a better way to evaluate the achieved yaw offset? For the South campaign, for instance, T1 is standing behind the complex terrain and could experience different wind conditions than the controlled T4 & downstream T3 turbines. Same might hold for T5, although not clearly visible in the terrain figure.

- page 6, line 16: missing closing bracket here

- page 11, line 7-12: do I understand correctly that for the South campaign the wind direction is obtained by averaging the measurements from the sodar, T1 and T5, but that the reference power is based only on T1, as T5 introduces too much scatter? But why is T5 then used for wind direction measurement in the South campaign?

- page 13, line 6: In Kanev (2019) no direct control over yaw is done, only dynamic generation of the yaw setpoint. The yaw controller remains unmodified.

- page 15, line 4: How was pP=1.9 chosen. I remember having seen lower values in some of your earlier field studies.

- page 16, line 8: could you please explain more clearly the difference between the wake losses in the left plots and those in the right plots in Figure 11? Which reference turbines are used in each case?

---

## Author Comment (AC1) · 28 Apr 2020

We thank the reviewers for their constructive feedback. Please see the attached response to reviewers where we document our improvements to the paper to accommodate reviewer feedback.

Please also note the supplement to this comment: https://www.wind-energ-sci-discuss.net/wes-2019-104/wes-2019-104-AC1-supplement.pdf

---

## Editor Comment (EC1) · Carlo L. Bottasso (Editor) · 29 Apr 2020

Dear Authors,

please go ahead and submit a revised version of your paper.

Kind regards Carlo Bottasso
* * *

---

## Author Response (AR1)

REVIEWER 1

This paper reports on a field test campaign for wake steering. Such field tests are vital to increase confidence in the usefulness of wake steering, while being difficult to design, and producing results which can be difficult to analyse conclusively. This campaign appears to have been well designed, and to have run for long enough to generate enough data to allow useful conclusions to be drawn. The paper presents a very clear description of the tests, of the analysis methods used, and the results obtained from the analysis, and is a valuable addition to the literature on this topic.

Thank you very much for these comments, it is greatly appreciated.

Although the paper is acceptable as it is, I have three minor comments to consider in any revision:

Section 6, page 11, line 10: the justification for the decision not to use T5 data in the South campaign is that the results are noisier (for credible reasons). It is important in any such analysis to demonstrate every effort to avoid unconscious bias that might result from such decisions, so it would be helpful to present some evidence to justify this decision.

This is a very fair comment and we do absolutely want to avoid biasing the results.

To answer this question, I went back through the data to re-explore the issues which lead to this decision.  The original decision was made in analyzing the phase 1 data, where a number of anomalous power variations in T5 were found.  Some of these behaviors were connected to un-noted changes in control state not flagged in the data (either related to curtailment or temperature protections).  In the original paper, the issue was avoided by using a "synthetic" power computed from the SODAR.   In phase 2 (which entirely includes the north campaign) we reduced these issues by working with the farm owner on better understanding and excluding the turbines from curtailment and other non-nominal power modes, and developing filters to catch those events which still occurred.  However, the south campaign, as discussed in this paper, includes the original phase, and so I decided to still exclude T5, but moved the reference to T1, instead of the SODAR because it updates more frequently and showed good sharpness in predicting the region of wake loss in the baseline case.  This seemed a reasonable decision, also because I was still assuming the difficult terrain to the south-east, and that in wake steering, T4 is steering it's wake towards T5 (although to be fair, it can be said that steering is directing wakes similarly away from T1), all made further prudent reasons to continue to exclude T5.

But in considering the reviewers comments, it's not obvious that these issues pose an insurmountable issue considering the controller is toggled, in other words, any issue should be in each band, and cancel out in comparison.   And also it could have lead to an inadvertent bias addition by allowing the references to be different between the two cases.
To test this I re-computed several of the figures from the current paper using the average of T5 and T1, rather than just T1, which I show here:

|  | Using T1 Reference | Using mean(T1,T5) ref |
|---|---|---|
| T3 Energy ratio |
[Figure]
 | |
| T3 Change in energy | | |
| T4 + T3 Energy Ratio | | |
| T4 + T3 Change in Energy | | |
| | | |
| | | |

In reviewing these results, the impact on the results of including T5 in the reference is not very big in a relative sense, however, one difference is that the wake losses appear a bit deeper. Perhaps a more important difference is that the energy ratio is not re-converging to 1.0 in the right half of the energy ratio figures as would be expected.  This result implies T5 is producing more energy than T3 and T4 even outside of the area of wake losses, which is most likely driven by the complex terrain over which the inflow arrives from the south.   This difference is not too impactful in terms of the relative energy gains, but it would affect the overall reduction in wake loss calculations because it generates apparent wake losses by not re-converging to 1.0.   For this reason, we prefer to keep the analysis as is, while recording this response in the author's response to reviewers, which is kept in the public record.

Same page, line 12: typo: 'resimulations'.

This is corrected thank you

Section 7.1, page 16, last line: the 'overall' figures for wake loss reduction in Table 2 are described as 'across all wind directions'. Is this an average across the bins, or an average weighted by the number of points in each bin?

Is is an average across bins, weighted by points per bin to ensure balance in wind distribution between control and baseline, the text is updated to make this clear

REVIEWER 2

Summary This paper is well written with substantial contributions. Specifically, it is of value to provide full scale tests that prove gains from using wind farm control. Publication is therefore suggested after minor revision.

Thank you very much for these comments.

Specific comments

1. P7 Figure 5 caption. "Colored bands are used to indicate where a yaw offset is achieved intentionally (light blue) and unintentionally (orange)."
    a. The notion of intentionally versus unintentionally is a bit confusing.
    b. When the active yaw controller yaws the turbine it is intended other wise the controller would have been designed to do something else?
    c. However, the measured combination of yaw, wind direction and speed is not at the ideal (steady state) values. This is of course not what is preferred/intended but just a consequence of actuator dynamics, limits, uncertainty etc. which are standard in any control task?
    d. Why call it unintended when there is a deviation between measured yaw and ideal yaw and the ideal is zero yaw but intended when there is an error but the ideal yaw is positive?

The reviewer is correct that there will always be a deviation between the ideal and realized. We should have been more clear that this discussion was targeted at a statistically achieved offset, so we revise the text as follows:

*The region is subdivided into a light blue region that indicates that yaw offset is applied and desired, whereas the orange region indicates that the yaw offset is achieved unintentionally. It is important to note that practically there is always some error between targeted and achieved yaw offset, so saying an offset is achieved in a given region, we refer that the mean offset is not zero.*

2. 2. P9 "Specifically, the TI is selected to provide a close match between the baseline wake losses in FLORIS to those measured in the field." Isn't it strange to use the TI to math wake losses? Shouldn't you use the field measure TI and then perhaps adjust wake parameters to match wake losses? (Now I see that the explanation is partly in the next sentence.)

We agree is it not an ideal solution, but as you note have identified the issue and highlight it for future work in the next paragraph. We hope this is sufficient for this paper.

3. 1 3. P10 "secondary steering (see Fleming et al. (2018b))" Please explain what that is?

This sentence is revised to:

secondary steering (secondary steering is the name for the effect described in Fleming et al. (2018b) where a steered-wake, interacting with a non-steered apparently induces steering into the non-steered wake)

4. 4. P 13 "The optimal gain (the expected gain if desired offset is always achieved) is higher than the realized gain."
   a. Will there be power gain results for the combination of the controlled turbine and the one in wake? (I will probably know when I have nished the paper)
   b. The problem of lower power gains compared to ideal static calculations do to the dynamics has been pointed to long back Knudsen et al. [2015]

Thank you for pointing this out, we now make reference to this known result in the updated paragraph:

*The optimal gain (the expected gain if desired offset is always achieved) is higher than the realized gain. As stated, this field campaign uses a first-pass lookup-table method to offset the vane signal provided to the usual yaw controller. We believe the optimal performance can be more nearly achieved by dynamic setting of the yaw setpoing (as in Kanev (2019)), accounting for yaw control*
*limitations in design (Simley et al. (2019a)), and by improving measurement of wind direction (for instance, using information sharing between turbines (Annoni et al. (2019)). However, realizing perfectly the optimal results is not possible as this would imply excessive yawing and perfect information of wind direction.Knudsen et al. (2014)*

5. 5. Figure 7 "this represents the ratio of energy produced by T3 with respect to unwaked reference turbines."
   a. If the controller were toggle on/of e.g. every 5 minutes you could compare with the same turbine in order not to risk differences between locations. 6.

After some discussion, we had chosen toggling the controller every hour, because we observe it takes a few minutes to be sure the yaw controller has responded to the offset, and then an additional few minutes to be sure the new wake properties have propogated fully downstream even in low-wind speed conditions. This implies that the first 5 minutes must be removed after each toggling, and also each toggling incurs additional yaw activity as the regime changes. For these reasons we selected 1 hour toggling

6. P21 "We nd this to be a very exciting result, as we believe that there are still more opportunities for improved performance for the next generation of wind farm controllers to approach higher percentages of the static optimum."
   a. As pointed out before we can not expect to reach the static optimum Knudsen et al. [2015]. 7

We agree, the revised response to comment (4) hopefully addresses this and we only mean that rather than getting 50% of the optimum, perhaps 67% is possible, or something in that line.

7. . P21 "Obtaining better knowledge of the inow conditions will also improve performance".
   a. A candidate method for this is model based state estimation as discussed in Doekemeijer et al. [2018].

Thank you for this suggestion, the revised text now includes this a possible method:

*Obtaining better knowledge of the inflow conditions will also improve performance. The consensus control algorithm of Annoni et al. (2019) provides a means for turbines to cooperate when estimating the wind flow in real time. This estimated consensus wind field can include spatial filtering and even preview, which could be of much use to the typically slow yaw controller. Other possibilities include incorporating the direct measurement of the inflow itself in the controls (Raach et al., 2019). An additional possibility is through online model estimation.Doekemeijer et al. (2017)*

REVIEWER 3

The paper is very well written and clear (as usual). It provides the results from the second phase of field trials with wake redirection. These results reconfirm and strengthen earlier findings (by the authors) indicating that wake redirection control has the potential to increase the power production of a wind farm. The paper provides very valuable insights into the operation of wake steering in the field and indicates possible improvement possibilities for the future.

Thank you very much for these comments!

page 2, line 19-21: please mention if the results in Howland (2019) are in line with yours, and explain any differences between their experimental setup and approach and yours. –

Text is expanded to:

*Since the first paper, an additional publication documenting a trial of wake steering at a commercial wind farm was published. (Howland, 2019) implemented a wake steering controller on an array of six turbines at a commercial wind farm and observed gains in power production for the waked cases tested. (Howland, 2019) differs from the current study in that the yaw offset angle is fixed (rather than controlled via lookup) and applied to multiple turbines, rather than a single control turbine. However, we believe the resultant gains are consistent with the current study.*

page 6, line 5: why is averaging the wind directions considered a better way to evaluate the achieved yaw offset? For the South campaign, for instance, T1 is standing behind the complex terrain and could experience different wind conditions than the controlled T4 & downstream T3 turbines. Same might hold for T5, although not clearly visible in the terrain figure. –

You're right, the reason for this change was not addressed previously, so we have added some motivation for this change:

*Fig. 5 summarizes all of the yaw offset data by wind speed and direction for both campaigns and both phases observed over the course of the campaign. The targeted offset is shown in black, whereas the achieved offset is shown in magenta. Note the achieved offset is calculated with respect to the reference wind direction (not the wind direction measured by the turbine itself as this could be affected by the yawing). The reference wind direction for the South Campaign described in the part 1 study was provided by the south sodar. However, in the present work, we moved to an average of measurements. For the North Campaign, this is the average of the wind direction measurements made by the lidar, as well as T1 and T5's wind direction measurement computed using the nacelle vane and measurement of yaw heading. For the South Campaign, this is T1, T5, and the south sodar averaged. We prefer this average approach for several reasons. First, by adding the turbine measurements, which are updated at 1Hz rather than the 10-minute average of the sodar, we include finer refinement in time. Secondly, the spatial separation of the measurements provides a wider look at the direction of the inflow which transports the wake.*

page 6, line 16: missing closing bracket here –

This is fixed now, thank you

page 11, line 7-12: do I understand correctly that for the South campaign the wind direction is obtained by averaging the measurements from the sodar, T1 and T5, but that the reference power is based only on T1, as T5 introduces too much scatter? But why is T5 then used for wind direction measurement in the South campaign? –

You are correct, please see the response to a similar question posed by reviewer 1. The decision to not include T5 in the power reference for the south campaign stemmed that certain anomalies in power of T5, but not wind direction, for which including T1 and T5 enables measuring wind direction on either side of T4.

 page 13, line 6: In Kanev (2019) no direct control over yaw is done, only dynamic generation of the yaw setpoint. The yaw controller remains unmodified. –

Thank you for catching this, we revised the description to:
*We believe the optimal performance can be more nearly achieved by dynamic generation of the yaw setpoint*

page 15, line 4: How was pP=1.9 chosen. I remember having seen lower values in some of your earlier field studies. –

This line changed to:
*In this work pP is set to 1.9 based on estimation from previous work (Damini, 2019) on a similar turbine.*

This coefficient for now we estimate turbine-by-turbine but haven't found a rule of thumb for what it will be. In our study with Envision, both SOWFA and field data were consistent on an estimate of 1.44. In SOWFA-based studies of the (model only) NREL 5MW turbine, a coefficient of 1.88 is found to be the best fit. 1.9 has been generally suitable to this current turbine.

- page 16, line 8: could you please explain more clearly the difference between the wake losses in the left plots and those in the right plots in Figure 11? Which reference turbines are used in each case?

Expanded figure caption:

*Wake losses computed per wind direction bin for the North and South campaigns. The wake losses in the left column are computed for the downstream T3 only, and then in the right, the total wakes losses of the combined upstream and downstream power for the respective campaign. In both cases, the references are as before, T1 and T5 for the north campaign, and T1 for the south campaign.*